# Commercial fishing amplifies impacts of increasing temperature on predator-prey interactions in marine ecosystems

Amy L. Shurety [1,2] ✉, Murray S. A. Thompson [2], Elena Couce[2], Tom C. Cameron [1] & Eoin J. O'Gorman [1]

Predator-prey interactions determine food web structure, energy flux, and ecosystem stability. Increasing temperatures and commercial fishing both alter body size distributions that underpin predator-prey interactions, but empirical evidence of their individual and combined effects is limited. We study how the predator to prey body mass ratio (PPMR) changes as a function of temperature and fishing effort in over 50,000 predator stomachs collected across the Northeast Atlantic over 35 years. PPMR increases with temperature, an effect that is exacerbated by greater fishing effort, driven by intraspecific decreases in prey body mass in heavily fished areas. To compensate for smaller prey (both within and across species) in warmer waters and areas of high fishing, predators target the largest prey available to them, but this is insufficient to alter the community-wide increase in PPMR. Higher PPMR is associated with weaker trophic interactions that dampen strong oscillatory dynamics but could also reduce energy transfer efficiency within ecosystems, both of which can affect ecosystem stability. These results could help underpin ecosystem-based management and sustainable fisheries by providing estimates of how future climate warming might interact with fishing to affect energy flux through marine food webs.

Marine food webs are highly size structured, consisting of many small organisms and few large organisms[1,2]. Trophic interactions tend to involve larger consumers eating smaller resources due to gape limitation and the lower risk of injury or wasted energy[3–5]. The size-structuring of marine food webs has given rise to their historical conceptualisation as trophic pyramids where the vast majority of biomass and productivity is concentrated at lower trophic levels made up of smaller organisms (e.g., plankton, small fish) with decreasing biomass and productivity at higher trophic levels that consist of larger intermediate and top predators (e.g., large fish, marine mammals)[6]. Thus, body mass is considered a master trait due to the constraints it imposes on trophic relations and is a useful tool for gaining insight into and modelling trophic interactions[2,7]. For instance, the relative sizes of predators and their prey measured via predator-prey mass ratios (PPMR) are a key constraint on how energy flows through ecosystems[8–10].

Predator-prey interactions act as highways of energy flow, creating avenues whereby the direct and indirect effects of species can propagate throughout entire ecosystems[11–15]. Trophic interactions are vulnerable to global environmental change[16,17], but the intricacies of the response are less well known[18–20]. Marine ecosystems exhibit large natural variability in temperature due to latitude, seasonality, and depth, but climate change is causing relatively rapid increases in sea surface temperatures (SST) beyond this natural variability[21,22]. For example, SST in the Northeast Atlantic has increased by as much as 0.5 °C in the last century[23], which has been linked to changes in biodiversity and food web structure[16,24–26]. Warmer waters are known to favour smaller species due to temperature dependencies of

[1]School of Life Sciences, University of Essex, Colchester, UK. [2]Centre for Environment, Fisheries and Aquaculture Science (Cefas), Lowestoft Laboratory, Lowestoft, UK. ✉ e-mail: amy.shurety@essex.ac.uk

distribution, physiology, and productivity[27,28]. Decreasing body mass is recognised as a ubiquitous response to increasing temperature, which can be due to physiological drivers such as increased metabolic demands keeping species smaller and the age at maturity lower[29,30], or through changes in community composition that favour smaller species[28,31].

Commercial fishing is a global anthropogenic pressure ubiquitous in marine ecosystems, particularly in the Northeast Atlantic, where the biomass of many fish species has declined dramatically over the past century[19,32,33]. Similar to the observed impacts of increasing temperatures, fishing also results in an increase in the relative abundance of smaller species, as commercial fishing practices are well known to target larger fish due to their commercial value[24,34]. By selecting for larger fish, commercial fisheries also impact the age-structure of fish communities[35,36]. Commercial fishing has a heterogeneous distribution in space and time and could therefore lead to complex effects on trophic interactions when combined with changes in temperature. For example, the combination of commercial fishing and increasing temperature has been shown to alter the recruitment, abundance, distribution, body condition, prey availability, and spawning success of marine organisms[26,37,38]. Despite the growing body of evidence highlighting the impacts of both commercial fishing and temperature on the size structuring of marine food webs, few studies have investigated the potential for complex interacting effects of these drivers on PPMR. It is important to address this because current models that forecast fish stocks under climate change use a fixed PPMR[39], not accounting for the physiological and compositional plasticity of species when exposed to environmental stressors.

To improve our predictive capacity, it is important to first understand the historical variation in PPMR across large-scale gradients of temperature and commercial fishing pressure. PPMR is an excellent predictor of the identity and strength of predator-prey interactions and thus for quantifying ecosystem energy flux[3,40]. Physiological and compositional changes, such as decreasing body mass and fluctuating species richness and prey availability, driven by increasing temperature and targeted fishing practices, could alter PPMR, which in turn has the potential to rewire whole food webs[41,42] and disrupt ecosystem functioning and stability[1,9,26]. Altered PPMR at the community-level is thus likely to be underpinned by disproportionate changes in the relative size of predators and their prey driven either by physiological mechanisms, such as altered metabolism or growth leading to a systematic reduction in body size of many species in the community (i.e. intraspecific processes), or compositional mechanisms, such as systematic changes in the number of small or large species within the community (i.e. interspecific processes). Disentangling the relative importance of these mechanisms is key to understanding and forecasting the effects of temperature and commercial fishing on the size-structure of trophic interactions.

Here, we employ a space-for-time substitution to quantify PPMR across spatial gradients of temperature and fishing effort, which could help predict how energy fluxes in the Northeast Atlantic may respond to future global change. Given that PPMR has a negative relationship with body mass[1,43,44], a reduction in the mean body size of the community at higher temperatures and/or commercial fishing pressure should lead to an increase in PPMR since smaller predators consume relatively smaller prey. We test the following hypotheses: (1) PPMR is positively related to temperature and fishing; (2) PPMR values are highest where temperature and fishing are both high; (3) increases in PPMR are related to both physiological processes (e.g., altered metabolic rate, which may underpin changes in body mass) and compositional processes (changes in taxonomic composition). The central goal of our study was to empirically quantify change in PPMR along a temperature gradient and test how this was affected by commercial fishing effort. Given how critical PPMR is in determining energy flux through marine food webs and their stability[10,19,45], this understanding

could be critical for improving projections of fish stocks and optimising multi-species fisheries management under future scenarios of climate change and commercial fishing.

## Results

### Effects of increasing temperature and fishing effort on PPMR

Across the Northeast Atlantic, there was a significant increase in PPMR with increasing temperature ($t_{64764} = 23.29$; $p < 0.001$, $R^2 = 0.45$; Fig. 1a, b). This suggests that predators and their prey diverged in size at higher temperatures (Fig. 1b). PPMR was predicted to increase by 30% across the temperature gradient, or 1.8% per 1 °C. Latitude was a major contributor to the effect of temperature on PPMR, with less contribution of season and water column depth (Fig. S1). The increase in PPMR was amplified in areas with more commercial fishing, with a significant interactive effect of temperature and fishing effort on PPMR ($t_{62436} = 2.99$, $p = 0.003$, $R^2 = 0.48$, Fig. 1c, d). PPMR increased at both high and low fishing effort, but the increase in PPMR with increasing temperature was more pronounced in areas that were more heavily fished (Fig. 1d). In areas of low commercial fishing, PPMR was predicted to increase by 22% across the temperature gradient, or 1.3% per 1 °C, whereas in areas of high commercial fishing, PPMR was predicted to increase by 82% across the temperature gradient, or 4.8% per 1 °C.

### Effects of only temperature on body mass, prey count, and prey richness

There was no significant effect of temperature on the mean body mass of the predators sampled ($t_{6660} = 1.93$, $p = 0.053$, Fig. 2a), i.e., using the larger dataset without fishing effort data included. Mean predator body mass was also relatively consistent at the species level across the entire temperature gradient, with no significant effect of temperature on 82% of predator species, accounting for 99% of predator biomass (Table 1, Table S1). In contrast, there was a significant decrease in the mean body mass of prey individuals with increasing temperature ($t_{64764} = -7.58$, $p < 0.001$, $R^2 = 0.83$, Fig. 2b), indicating that reductions in prey body mass were a primary driver of the increase in PPMR with increasing temperature. Note that we refer to "prey" here as any organism found in the diet of the fish predators, even if those species could be predators themselves. At the species level, there was a decrease in mean individual prey body mass with increasing temperature for 58% of prey species, accounting for 77% of prey biomass (Table 1, Table S2), indicating that intraspecific responses to increasing temperature played a major role in determining the observed changes in PPMR.

The prey count ($t_{6660} = -4.91$, $p < 0.001$, $R^2 = 0.64$, Fig. 2c) and prey species richness ($t_{6660} = -6.42$, $p < 0.001$, $R^2 = 0.82$, Fig. 2d) within predator stomachs significantly increased with increasing temperature. There was a significant interaction between prey size class and temperature for both prey count ($t_{6660} = 13.07$, $p = <0.001$, $R^2 = 0.64$, Fig. 2c) and prey species richness ($t_{6660} = 27.43$, $p < 0.001$, $R^2 = 0.82$, Fig. 2d). Here, predators increasingly targeted larger prey from a bigger pool of species as temperature increased. However, the nMDS analysis did not indicate any strong relationship between the size of the prey taxa consumed and mean temperature (Fig. 2e), indicating no clear evidence for compositional changes contributing to the decrease in mean prey body mass with increasing temperature.

### Interactive effects of temperature and fishing effort on body mass, prey count, and prey richness

The consistency of sampled predator body mass across the temperature gradient was maintained despite additional impacts of commercial fishing, with no significant interactive effect of temperature and fishing effort ($t_{6085} = -0.71$, $p = 0.478$, Fig. 3a). There was a significant interactive effect of temperature and commercial fishing on individual prey body mass ($t_{62436} = -9.92$, $p < 0.001$, $R^2 = 0.83$; Fig. 3b), however, with an increase in prey body mass at low fishing effort and a decline in

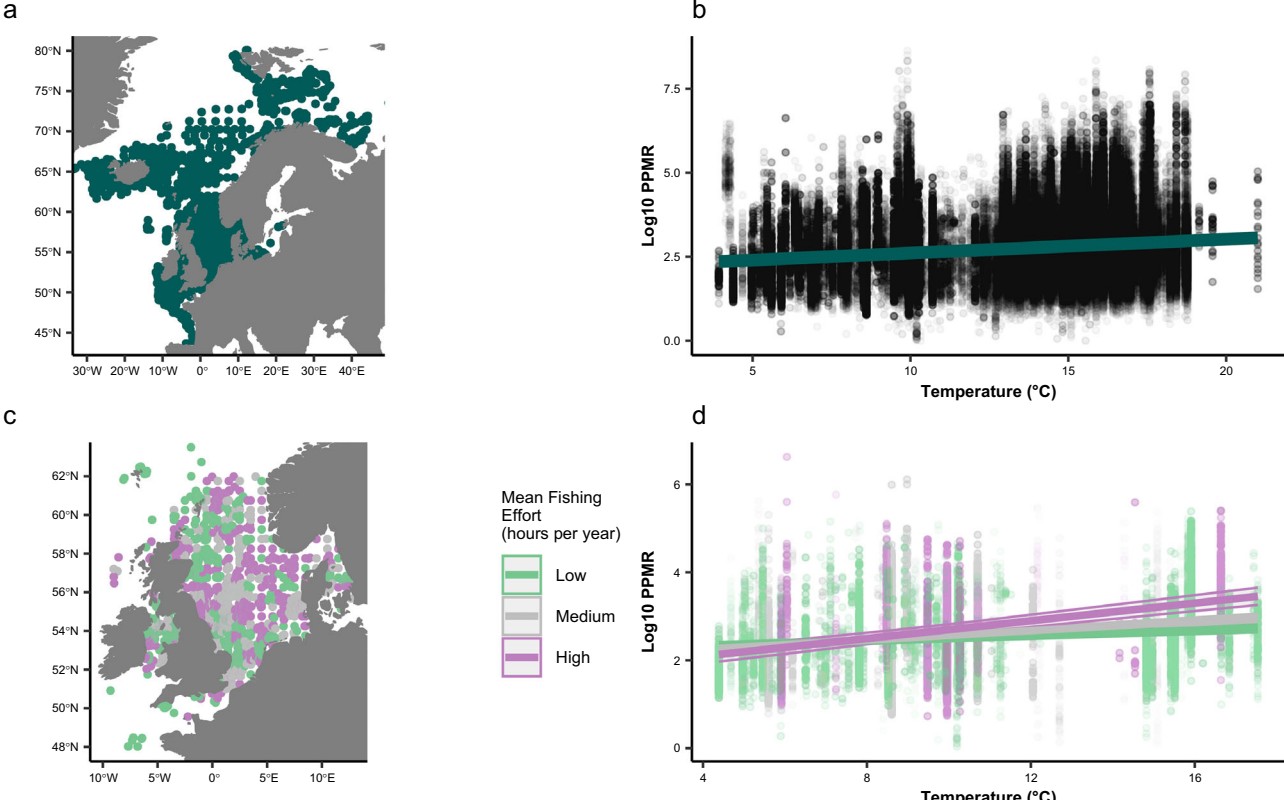

**Fig. 1 | They key findings of temperature and commercial fishing driven changes in PPMR across the Northeast Atlantic. a** A map of the Northeast Atlantic study region illustrating all sites (points) that were sampled over the period 1981 to 2016. **b** the relationship between temperature (°C) and $\log_{10}$ predator prey body mass ratio (PPMR) with the fitted linear mixed effects model ($y = 2.2 + 0.041x$). Temperature refers to the mean SST (°C). **c** A map illustrating the sampled sites (points) that had corresponding commercial fishing data available. **d** The inter-active effect of commercial fishing on the relationship between temperature and PPMR. Commercial fishing effort was analysed as a continuous variable, but for the purposes of visualisation, low (45–566 h per year), medium (576–878 h per year), and high (903–3479 h per year) levels of fishing effort are indicated in the figure. **d:** low fishing effort: $y = 2.2 + 0.03x$; medium fishing effort: $y = 2 + 0.053x$ and high fishing effort: $y = 1.7 + 0.1x$. The maps were designed in R Software using the maps Package87. The maps were created using the maps R package[89]. Source Data are provided as a Source Data file.

prey body mass at high fishing effort (Fig. 3b). Fishing effort also significantly altered the positive relationship between prey count and temperature ($t_{6085} = 15.0$, $p < 0.001$, $R^2 = 0.79$, Fig. 3c), with a reduction in the number of prey of all sizes consumed with increasing temperature at low fishing effort, but an increase in the number of prey of all sizes consumed with increasing temperature at high fishing effort. There was a significant interactive effect of temperature and commercial fishing on prey species richness ($t_{6085} = 5.19$, $p < 0.001$, $R^2 = 0.84$, Fig. 3d). Here, there was a greater increase in prey species richness with temperature as prey size class increased and as fishing effort increased. The nMDS plot showed a tendency for larger prey taxa to be found in areas of increased commercial fishing, although mean fishing effort only accounted for 4% of the variation in body size ($R^2 = 0.04$, Fig. 3e). Mean prey body mass was also not strongly associated with mean temperature or fishing effort.

## Discussion

We provide empirical evidence that commercial fishing amplifies the increase in PPMR with increasing temperature in the Northeast Atlantic (Fig. 1d), as sampled predators in warmer waters with higher fishing effort typically consumed the smallest prey relative to their body size. This suggests that increased water temperature within heavily fished ecosystems of the Northeast Atlantic could cause predator and prey body mass to diverge from one another. A larger community-level PPMR typically results in a less efficient flow of energy through food webs[46], which could reduce the persistence of apex predators and

overall system stability[47]. Quantifying the effects of multiple stressors on predator-prey interactions across large spatiotemporal gradients is thus crucial to gain insight into how ecosystems could respond to global change and improve ecosystem-based management. Our space-for-time substitution provides insights into past variation and potential adaptation of trophic interactions to environmental drivers such as higher temperatures, given the long timescales over which ecosystems have been exposed to these conditions, i.e., organisms may be thermally adapted after many generations at higher temperatures, which is not possible to study in short-term experiments involving acute temperature exposures.

The observed reduction in the mean body mass of the prey community at higher temperatures (Fig. 2b) was underpinned by overall intraspecific reductions in body size (Table 1). This indicates a physiological response to increasing temperature, which can occur through changes in metabolism, ontogeny, and thermoregulation, all of which can contribute to the evolution and persistence of smaller-sized species in warmer waters. Metabolic rates are known to rise with temperature and, as a result, increase the energetic requirements of organisms[29]. This is exacerbated by reduced oxygen in warmer waters further constraining energetic demands which can be more easily maintained by smaller individuals[28,48–50]. The increase in metabolism causes species to have faster growth rates and shorter generation times, which can also contribute to decreased body mass if species mature and reproduce quicker in warmer areas[51,52] (though note there are exceptions to this general rule[53]). Furthermore, when close to the

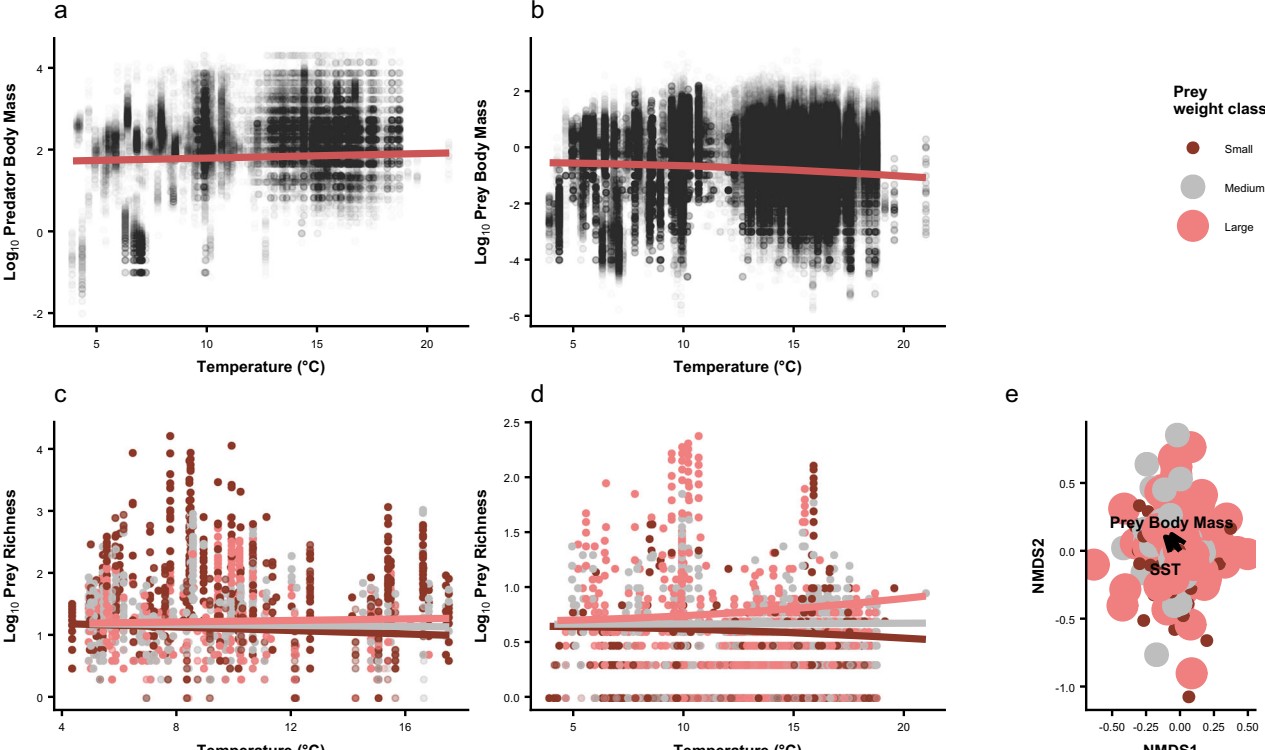

**Fig. 2 | The effect of temperature (°C) on underlying predator and prey variables including. a** $\log_{10}$ predator body mass (g) ($y = 2.3 + 0.0029x$), **b** $\log_{10}$ prey body mass (g) ($y = -0.53 - 0.0012x$), **c** $\log_{10}$ prey count (number of individuals per stomach) and **d** $\log_{10}$ species richness (number of species per stomach) in the Northeast Atlantic between 1981 and 2016. The results of (**c**) $\log_{10}$ prey count and (**d**) $\log_{10}$ species richness were further categorised into prey weight classes by small (0–0.072 g); medium (0.072–1.25 g) and large (1.25–2137 g) prey body mass (**c**: small species: $y = 1.2 - 0.009x$; large species: $y = 1.1 + 0.021x$; **d**: small species: $y = 0.65 + 2.6 \times 10^{-17}x - 0.00028x^2$; large species: $y = 0.69 + 4 \times 10^{-17} + 0.00053x^2$). **e** nMDS illustrating the weak effect of temperature on the size of prey taxa in the community, with short temperature and body mass vectors that are orthogonal to one another. The colour and size of the points represent the size classes of the prey taxa. Source Data are provided as a Source Data file.

upper limits of their thermal ranges, smaller individuals thermoregulate more efficiently than larger individuals as they are better able to lose excess heat due to their large surface area-to-volume ratio[54]. Our results show that individual body mass within the majority of sampled prey species decreased with increasing temperature (Table S2). Other studies have also found temperature-driven decreases in the size of individual Northeast Atlantic species such as plaice, *Pleuronectes platessa*[50] and Atlantic cod, *Gadus morhua*[55], due to the temperature-dependent physiology of marine species.

Changes in PPMR could also be a consequence of altered community composition[42,56–60], however, no clear evidence of size-based compositional changes behind the increase in PPMR were observed in our study (Fig. 2e). In other words, while the taxonomic composition of prey species may have changed from colder to warmer waters, the number of small and large taxa was still similar. Despite the similar size composition of prey species across the temperature gradient, there was a community-wide intraspecific decrease in prey body mass at higher temperatures and fishing effort (Table S2), i.e., driven by reductions in size within rather than across species. The sampled predators responded to the intraspecific decrease in prey body mass by selecting the largest individuals available to them (Fig. 2c) and a greater diversity of large prey (Fig. 2d), which could be a result of optimal foraging in order to maintain energetic requirements[61]. Nevertheless, the overall reduction in average size of the prey community meant that PPMR increased with both temperature and fishing effort (Fig. 1b, d). Changes in the abundance and selectivity of predators could also reflect changes in the environmental abundance and distribution of prey. To test this would require information on the quantity of prey in the environment as well as in the stomachs of

**Table 1 | The percentage of each predator and prey species, their respective biomasses, and the average individual body size response to increasing temperature using the larger dataset without fishing effort data included**

|           | Predator species | Predator biomass | Prey species | Prey biomass |
|-----------|------------------|------------------|--------------|--------------|
| No change | 81.6             | 99.4             | 32.3         | 23.0         |
| Increase  | 5.3              | 0.1              | 14.0         | 0.2          |
| Decrease  | 13.2             | 0.5              | 58.2         | 76.8         |

predators, but these data are not yet routinely collected at the scale needed for this study.

Commercial fishing is known to be a strong driver behind the size-structure within marine ecosystems, sometimes even more so than increasing temperature[62,63]. Here, we found that commercial fishing amplified the increase in PPMR across the temperature gradient of the Northeast Atlantic (Fig. 1d). In particular, commercial fishing magnified the reduction in the body mass of prey species with increasing temperature (Fig. 3b), which underpinned the increase in PPMR in warmer waters with more fishing. The long-term targeting of larger individuals by fisheries reduces the body mass of commercially valuable species[24,34,64–66]. For example, Atlantic mackerel (*Scomber scombrus*), haddock (*Melanogrammus aeglefinus*) and European plaice (*Pleuronectes platessa*) are all commercially important species within the prey community that exhibited a decline in body mass. This is supported by previous research within the Northeast Atlantic showing that the average body mass of mature mackerel declined by as much as

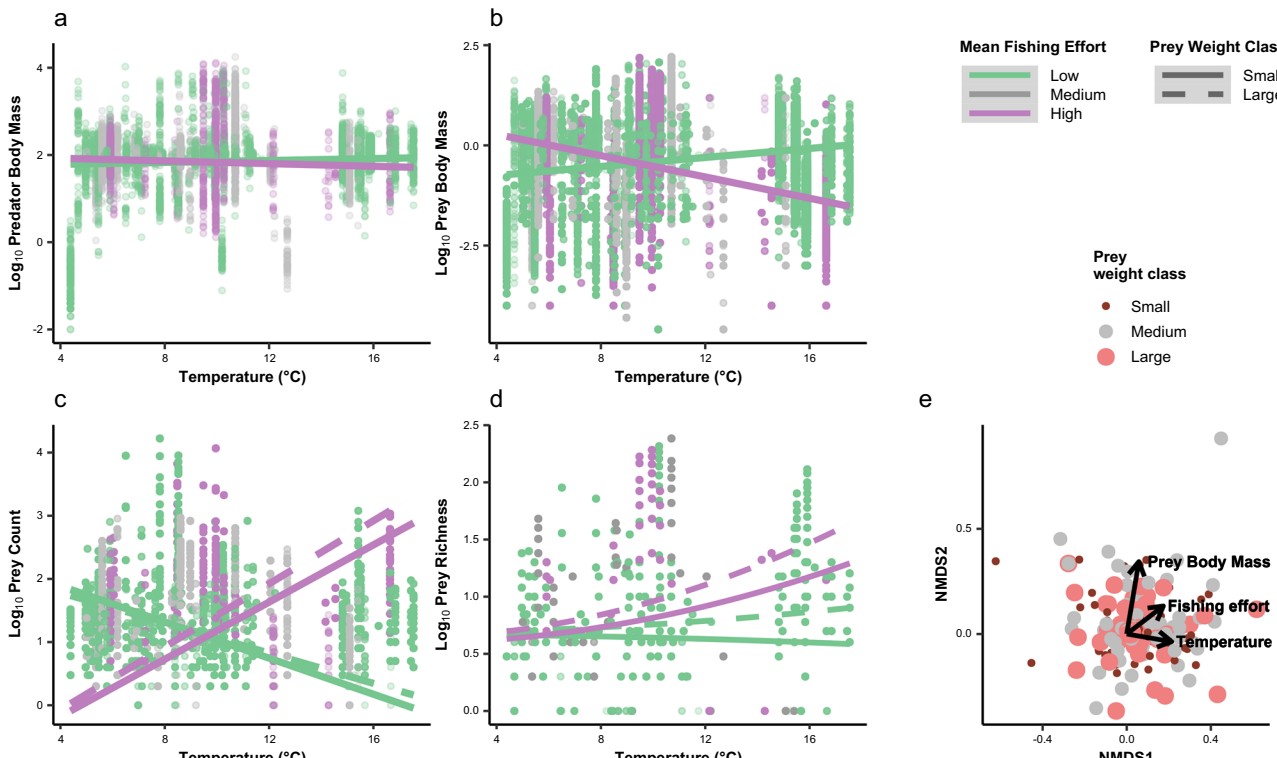

**Fig. 3 | The interactive effect of temperature (°C) and commercial fishing on underlying predator and prey variables, including. a** $\log_{10}$ predator body mass (g), **b** $\log_{10}$ prey body mass (g), **c** $\log_{10}$ prey count (number of individuals per stomach) and **d** $\log_{10}$ species richness (number of species per stomach) in the Northeast Atlantic between 1981 and 2016. Commercial fishing effort was analysed as a continuous variable, but for the purposes of visualisation, low (45–566 h per year), medium (576–878 h per year), and high (903–3479 h per year) levels of fishing effort are indicated in the figure. Trend lines based on the predicted values of the mixed effects model were plotted for low (**a**: $y = 1.6 + 0.026x$; **b**: $y = -1.1 + 0.0039x$) and high commercial fishing (**a**: $y = 1.8 + 0.0081x$; **b**: $y = 1.9 + 0.003x$). The results of (**c**) $\log_{10}$ prey count and (**d**) $\log_{10}$ species richness were further categorised into prey weight classes by small (0–0.072 g); medium (0.072–1.25 g), and large (1.25–2137 g) prey body mass. (**c**: low fishing effort and small species: $y = 2.5 - 0.14x$; high fishing effort and small species: $y = -1.1 + 0.23x$; low fishing effort and large species: $y = 2.3 - 0.12x$; high fishing effort and large species: $y = -1.1 + 0.25x$; **d**: low fishing effort and small species: $y = 0.67 - 4.3 \times 10^{-17}x - 0.00027x^2$; high fishing effort and small species: $y = 0.59 - 3.5 \times 10^{-17}x + 0.0023x^2$; low fishing effort and large species: $y = 0.69 - 2.3 \times 10^{-17}x + 7 \times 10^{-4}x^2$; high fishing effort and large species: $y = 0.63 - 5.8 \times 10^{-17}x + 0.0033x^2$). **e** nMDS illustrating the weak effects of temperature and fishing effort on the size of prey taxa, with temperature and body mass vectors orthogonal to one another and a weak correlation between body mass and commercial fishing effort ($R^2 = 0.04$). The colour and size of the points represent the size classes of the prey taxa. Source Data are provided as a Source Data file.

175 g per year from 1983 to 2013[67], a mature haddock by as much as 9% from 1970 to 2008[68], and the maximum body mass of plaice by 28% throughout the 20th century[69], with direct links to the intensification of commercial fishing. Smaller species have become increasingly abundant in fisheries landings due to the collapse of many larger commercial species[70,71], suggesting that the impacts of fisheries are not just found at higher trophic levels. Large-scale ecosystem degradation is occurring simultaneously through destructive fishing methods, such as trawling, that reduce habitat diversity and food availability, which could impact trophic interactions throughout the size-spectrum of a food web[72,73].

The sampled Northeast Atlantic predators in this study were seemingly more resilient to increasing temperature, with no change in body mass across the temperature gradient (Fig. 2a) or in response to fishing (Fig. 3a). The consistency in predator body mass could be due to sampling bias of the survey trawls targeting larger, commercially important fish species, i.e. smaller species not captured by the trawls may also be predators in the wider food web, and so the lack of temperature effect on predator body size is largely constrained to organisms in the 1 to 10,000 g size range. The trawls had no control over the prey composition included, i.e., since the prey in the stomach contents were selectively sampled through feeding by the predators.

Our original hypothesis that PPMR would increase with temperature in the Northeast Atlantic was underpinned by the expectation of widespread reductions in body mass in warmer waters (i.e., for both predators and prey). Given the observed negative relationship between PPMR and body size[1,43,44], a community with a smaller average body mass should thus have a larger PPMR. This hypothesis was only substantiated by a decrease in prey body mass (Fig. 2b), not also predator body mass as anticipated (Figs. 2a, 3a). Thus, the overall increase in PPMR with temperature was driven by a different mechanism than our expectation, i.e., contrasting effects of temperature on large and small organisms leading to a divergence in their size ratio. Differential effects of temperature on the body size of trophic groups have recently been described in coastal reef ecosystems, with a similar reduction in the size of smaller fish with increasing temperature and no change in larger piscivores[31]. This highlights the importance of further research to explore the underlying mechanisms to improve our understanding of how warming may affect different food web compartments and thus overall functioning and stability.

Increasing PPMR, as seen here across the temperature and fishing effort gradients of the Northeast Atlantic, is associated with a greater prevalence of weak interactions[41], which can help to buffer against the destabilising, oscillatory dynamics of strong trophic interactions[74]. Weak interactions are often associated with more generalist predator diets that hold more trophic redundancy, i.e., more pathways for

energy to flow through the food web[74,75]. Flexible foraging or generalised predation can also prevent the over-dominance or over-predation of particular prey species under different environmental conditions, helping to increase ecosystem stability[76] Case studies have shown that the impacts of species loss are more unpredictable in food webs that have lower redundancy[77]. On the other hand, weak per capita interactions can also result in species needing to consume more or larger prey to meet their energetic requirements[61], leading to stronger population-level interaction strengths. This was observed in the Northeast Atlantic, as sampled predators were found to be targeting more individuals and species of larger prey in warmer waters (Fig. 2c, d). This highlights how temperature can interact with factors like PPMR and the associated indirect changes to trophic interaction strengths to alter entire community size spectra[78]. Thus, the consequences of increasing PPMR for the persistence of species and overall energy flux through food webs are still uncertain, and are an area requiring urgent attention in future studies.

PPMR is a valuable metric to monitor ecosystem changes as it is a good predictor of trophic interactions, governs how energy flows through ecosystems[1,2,13], can estimate community size-spectrum[78], and as a result, is a fundamental input into size-based models of ecosystem dynamics such as the Allometric Diet Breadth Model[2,3]. In such models, PPMR is normally treated as a constant parameter[39] without accounting for systematic variability under different environmental conditions. Here, we show that PPMR should instead be considered as a dynamic parameter to account for the heterogeneity of body mass across environmental gradients. Future research could analyse how changes in PPMR propagate through entire food webs, e.g., via dynamical models, to further understand how energy stocks and fluxes are affected by such changes.

Our study suggests that impacts of climate change, such as increasing temperature, will be more pronounced in areas favoured by commercial fisheries, providing evidence to promote ecosystem-based management, especially in regions experiencing notable increases in temperature and high commercial fishing, to better maintain predator-prey interactions. This illustrates the complex nature of changing environmental gradients on marine food webs and highlights the importance of ecosystem-based management that considers the individual and synergistic impacts of multiple environmental change drivers. Thus, climate change policies and fisheries management should be integrated in order to make meaningful impacts when managing the trophic structure of marine ecosystems.

## Methods
The study relies on historical stomach-content records publicly available from the Cefas Data Hub[79], Copernicus[80–82] and the JRC Data catalogue[83]. No new animal sampling was undertaken, and therefore, ethical approval and informed consent were not required. Data were used in accordance with institutional and national regulations.

### Study region
The fish stomach survey data for this study were collected from 1981-2016 in an area of the Northeast Atlantic spanning 35° of latitude and 70° of longitude, incorporating the Bay of Biscay, Celtic Sea, North Sea, Norwegian Sea, and Greenland Sea (Fig. 1a). The geographic area has a temperate climate in the south and a polar climate to the north[7]. The region provides ecosystem goods and services to large populations across many countries in western Europe, including valuable commercial fish stocks, with mature oil and gas fields, rapidly developing offshore wind infrastructure and important carbon sinks[7,84,85]. The Northeast Atlantic thus experiences some of the strongest anthropogenic impacts globally[26,73,86]. Extensive and often coordinated international research has also been conducted, providing extensive datasets and sampling records[87].

### Stomach content dataset
Observations of PPMR were taken from the Dapstom stomach content database[79]. We utilised a total of 313,953 individual observations from 53,444 individual stomachs of 88 unique predator species. These observations were made on 1,862 different research hauls across the Northeast Atlantic (44° N to 79.5 ° N and 28.5 E° to 41.9° W) from 1981 to 2016. Predators were always identified to species level, with prey identified to the highest possible taxonomic level, i.e., species where possible, but often to family level. All prey species were considered (i.e., both fish and invertebrates). Fullness of stomach or level of digestibility were not considered, and so the estimates of body mass may be subject to some associated uncertainty. Additional variables taken from the database included the prey abundance per predator stomach (prey count), the geographical coordinates, and the year and month each sample was collected. These variables were included in the study as any change in PPMR can be driven by multiple and not mutually exclusive processes, e.g., changes in predator body size, prey body size, predator behaviour to select different-sized prey, or the behaviour of prey to avoid predation based on their body size.

Biomass-weighted PPMR was calculated for each individual predator using the following equation (1)[60]:

$$PPMR = \frac{M_i}{\frac{1}{n}\sum_{j=1}^{n} Mj}$$

where $M_i$ is the body mass of predator species $i$, $M_j$ is the body mass of prey taxon $j$, and $n$ is the total abundance of prey in the stomach. Wet weight (g) was defined as body mass.

### Sea-surface temperature dataset
Daily sea surface temperature (SST) data (°C) from both satellite and in situ observations were extracted from the Copernicus open access data repository[80–82]. Throughout the rest of the study, SST is simply referred to as temperature. The spatial resolution of the temperature data was 0.05° longitude x 0.05° latitude and covered every month from 1981 to 2016. The average temperature (°C) was calculated per month and matched to PPMR data sampled in the following month to account for any lag effects (i.e., if a predator's stomach contents were sampled in May 1992, then the corresponding temperature would be the monthly average of April 1992). Other environmental variables used for modelling purposes included salinity, chlorophyll (ug/l), and the average water column depth (m), which were taken from the ICES open-source data portal (ICES Data Portal, Dataset on Ocean Hydro-Chemistry, Extracted June 12, 2023. ICES, Copenhagen). These ecologically relevant variables were included as they could cause background variation in PPMR. The mean of each environmental variable was calculated for each month of every year (1981–2016) and matched to the following month of PPMR data. The longitude and latitude coordinates were to four decimal points. Due to spatio-temporal limitations in the environmental data layers available, 43% of PPMR observations did not have corresponding salinity or chlorophyll data.

### Commercial fishing effort dataset
We make use of the Scientific, Technical and Economic Committee for Fisheries (STECF) trawling effort dataset, available from the JRC Data catalogue, because of its extensive coverage in space and time, which corresponded with our PPMR observations (STECF, 2017). The STECF data provides annual fishing hours per ICES rectangle (0.5° latitude by 1° longitude) across areas of the Northeast Atlantic. We downloaded the data for the region 49.25° N to 63.25° N and 7.5° W to 12.5° E, covering the period 2002 to 2022. The data is a compilation of member state submissions in response to the Data Collection Framework (DCF) Fishing Effort Regimes Data Call in 2017[83]. The STECF dataset used in this study included data from Belgium, Denmark, the

Netherlands, the United Kingdom, France, Germany, and Sweden. Fishing effort was matched to the PPMR observations based on the year and the ICES rectangle in which the sampling took place. This resulted in 131,767 PPMR observations in the Northeast Atlantic from 2002–2016 with a corresponding measure of fishing effort in hours per year (Fig. 1c). For visualisation purposes, the fishing effort data were divided into three bins, each containing an equal number of observations, and the median of each bin was calculated for fitting regression lines to plots. An overview of the data construction can be found in Fig. S2.

## Statistical analysis

All statistical analyses were conducted using R v4.2.2[88]. A mixed effects model was used to test the relationship between temperature and either PPMR, predator body mass (g), prey body mass (g), prey abundance per predator stomach (prey count), or prey species richness per predator stomach. The response variables were $\log_{10}$ transformed to meet the assumptions of normality, homogeneity, and independence of residuals (Fig. S3). The model included the fixed effect of mean temperature (°C, continuous variable) and random effects for chlorophyll (ug/l, continuous), salinity (‰, continuous), depth (m, continuous), number of years since the start of the study period (continuous), ICES rectangle (categorical), season (categorical), predator species identity (categorical), and predator stomach identity (categorical).

The same five response variables (PPMR, predator body mass, prey body mass, prey count, and prey species richness) were then separately included in models containing main and interactive effects of mean temperature (°C, continuous variable) and mean commercial fishing effort (hours per year, continuous variable) for the subset of data that included information on fishing effort. The response variables were again $\log_{10}$ transformed to meet the assumptions of normality, homogeneity, and independence of residuals (Fig. S4). The random effect structure remained the same as above. For all models, linear and polynomial versions were compared using Akaike's Information Criterion (AIC), with the linear models found to be the best fit in seven out of ten comparisons (Table S3).

Lastly, nMDS ordination plots were constructed for both the full temperature dataset and the fishing effort data subset. Prey were aggregated into families for consistency and to help with model convergence, and plotted against mean temperature, prey body mass, and fishing effort vectors.

## Data availability

The data used within this study can be accessed through the University of Essex research open access data repository (A.L. Shurety, M.S.A. Thompson, E. Couce, T. Cameron and E.J. O'Gorman, Commercial fishing amplifies impacts of increasing temperature on predator-prey interactions in marine ecosystems. University of Essex Research Data Repository. https://doi.org/10.5526/ERDR-00000220. 2025). This study also made use of the DAPSTOM database[79] which can be found on the Cefas data hub (https://www.cefas.co.uk/data-and-publications/fish-stomach-records/), as well as sea surface temperature records[80–82] and fishing effort data[83]. Source data are provided with this paper.

## Code availability

The code used within this study is available via GitHub: https://github.com/amyshurety/PPMR_Atlantic_Publication/tree/main.

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

## Acknowledgements

We thank the researchers who constructed the data behind our analyses, especially Johan Pinnegar from the Centre for Environment, Fisheries and Aquaculture science (CEFAS). We acknowledge funding from CEFAS, the University of Essex, and NERC (NE/Y001184/1; NE/V016016/1; NE/V017039/1).

## Author contributions

All authors conceptualised and developed the research idea. M.S.A.T. and E.C. provided the data. A.L.S. performed the statistical analyses, which were reviewed by E.J.O., M.S.A.T., and E.C. The writing of the publication was led by A.L.S. and E.J.O. All authors, including T.C, prepared, reviewed and edited the final draufts.

## Competing interests

The authors declare no competing interests.
