## [Transparent Peer Review file · Nature Communications]

Commercial fishing amplifies impacts of increasing temperature on predator-prey interactions in marine ecosystems

Corresponding Author: Dr Eoin O'Gorman

Version 0:

Reviewer comments:

Reviewer #1

(Remarks to the Author)

I have very much enjoyed reading the manuscript addressing how the interaction between gradients in temperature and intensive fishery affect the predator-prey body-mass ratios of fish predators. The question targeted is highly relevant and considers the interplay between various anthropogenic stressors. The manuscript is well written and the methodological and statistical procedures are presented with all detail necessary to understand or replicate the study. Most of my initial points, after giving the manuscript a quick read, such as effects of spatial autocorrelation and non-linearity in the relationships have been covered in the method section. I was highly impressed with the rigor how this analysis has been carried out. Therefore, I have only very few points, and I hope that they are helpful.

P1: The study is based on a space for time substitution. The temperature gradient arises from large natural variability in temperature due to latitude, depth, and seasonality (line 49). I would encourage the authors to also test for the effects of the underlying variables (latitude, depth, seasonality). This is not meant to replace the statistical tests in the main manuscript but it would be great to see the results as a supplement to understand which of these factors is the main underlying driver.

P2: The authors have used mixed models with a variety of random effects. I was wondering why predator identity has not been used as a random effect. Of course, this can only be done for predators with sufficient replication across the temperature gradient but it would be important to show that these are effects of temperature not affected by any turnover in predator identity.

P3: I like how the results are presented in the figures. Nevertheless, I would like to get a better feel for the quantitative effects. I would like to see some numbers on what is the predicted average change in PPMR on the temperature gradient between five and twenty degrees Celsius.

P4: In Fig 1d, the interaction effect and the model predict that intensive fishery increases PPMR at high and decreases it at low temperatures. Is this reality or a model artefact? If this is a model artefact, a non-linear model might do a better job despite not being favored by AIC selection.

A minor point: line 103: I would not call a change in body mass a physiological effect.

I hope that these points are useful. As always, I have signed my review and I am happy to discuss these points if anything should be unclear.

Signed

Ulrich Brose

Reviewer #2

(Remarks to the Author)

The authors leverage an outstanding dataset, probably the most extensive of its kind globally, to investigate a key parameter of size-spectrum models: predator-prey mass ratios. This comprehensive data encompasses both temperature and fishing-pressure gradients, allowing for two potential drivers of fish body size variation in the region to be assessed.

Key results include:

There is a significant increase in PPMR with increasing temperature (regardless of fishing pressure).
The authors next state that there is an interaction effect between temperature and fishing pressure which amplifies the increase in PPMR with temperature.
The mean size of predators was not related to temperature (regardless of fishing pressure).
Temperature was significantly related to the mean prey size, and interrelated variables: prey count and prey species richness.

The above findings are based on mixed effects models of five response variables (i) $\log_{10}(\text{PPMR})$, (ii) predator body mass (g), (iii) prey body mass (g), (iv) prey abundance per predator stomach (prey count), and (v) prey species richness per predator stomach. The predictor variables included: (1) fixed effect of mean temperature ($^{\circ}\text{C}$), random slopes of (2) chlorophyll, (3) salinity, (4) depth, along with random intercepts of (5) stomach contents data source, (6) year since start of study period, (7) ICES rectangle, (8) predator species identity. These response variables were then re-tested in a model that included the interaction between temperature and commercial fishing effort. The authors state that polynomial and linear models were compared using AIC, and that linear models consistently outperformed polynomial alternatives.

While the authors use the well-established approach of linear mixed effects to study the space-for-time patterns of the above variables, there are a few areas that the authors could improve. Mostly these are related to demonstrating the validity of their results and careful interpretation.

The dataset they draw on is publicly available, but the code used for analyses to derive the results was not, this would be beneficial for others and would help with repeatability, transparency, interpretation, and for testing the generality of their results. Also which R package was used for these analyses?

Greater clarity of the data and statistical model structure - is each data point an individual predator-prey link? It is not always clear if the authors are referring to the mean individual across all species sampled or the species - often the term "individual species" is used, which is confusing and misleading given the random effect of species in the model. The flowchart in the supplement is helpful but would be improved with these additional details. Also, it should be explained in the methods what the later shown statistics are being used to test for. For example, Table S1 is provided to show that predator and prey body sizes do not significantly change with temperature - at the species level, again the nesting in the different tests is not fully clear and should be explicit. Table S1 seems to show predator species not prey as specified in the caption and text. In a size-based view of the world, predators are also prey depending on body size.

Predictor variables: Was fullness of stomach considered or level of digestibility? (level of digestibility can make prey sizes inaccurate). Were the prey types considered only fish (as is often the case) or all measured prey (e.g. invertebrates)?

The spatial pattern of high, medium, and low categories of fishing effort used for visualisation (Figure 1) seems almost random - how were these different levels of effort determined (were they based on quantiles or a cluster analysis or just arbitrarily chosen?).

It would be helpful to show the residual/diagnostic plots of the model in the supplement. The model predictions shown in Figure 3b at low temperatures and high fishing don't really have any data points, and this is the pattern that drives the interaction term. Appreciate that polynomials were tested, how was model selection carried out? Seeing diagnostics for the mixed effects models would help (e.g. recommendations here: <https://besjournals.onlinelibrary.wiley.com/doi/full/10.1111/2041-210X.12577>).

Also, please show results in supplement to support this statement: "For all models, linear and polynomial versions were compared using Akaike's Information Criterion (AIC), but the linear models were consistently found to be the best fit and thus are presented here."

Overall, while the paper is well-presented, with potentially wide-reaching findings, I find it difficult to assess the validity of these findings in the present version, and therefore whether commercial fishing is really amplifying the PPMR - temperature relationship, without these further details.

More specific comments

Line 15 - 17

The abstract is very well written, excepting this one line where it is unclear whether the same species are found at warm and colder waters, or whether the sentence is referring to different species. The last part of the sentence would be fine if the first bit of the sentence was clear.

Line 33

I would avoid starting sentences with 'their' as it is an ambiguous word, 'trophic' could be a better word in this case.

Line 34

The use of "lower risk". I think know what you mean, but this is very vague. I am not sure it would be clear to others, perhaps 'lower risk of injury or wasted energy'.

Line 40

"Super trait". Master is often used, but I wonder if you are choosing 'super' to avoid the use of 'master', which is up to you.

Line 42

Not sure 'moreover' is appropriate here (given the sentences leading up to it).

Line 50

This is where 'moreover' would be appropriate as it is sort of piling more onto the previous sentences. "However" would be appropriate if the previous sentences were contradictory, which they are not.

Line 51

"natural thermal ranges" of what? Please specify what the natural thermal ranges are of, otherwise it is vague.

Line 56 - 57

1) Not solely, it can also be brought about by species redistributions, see Cheung et al., 2013 and others,

and possibly through trophic guild-temperature trends, see

Coghlan AR, Blanchard JL, Wotherspoon S, Stuart-Smith RD, Edgar GJ, Barrett N, Audzijonyte A. Mean reef fish body size decreases towards warmer waters. Ecology Letters. 2024 Feb;27(2):e14375.

2) The TSR is only one mechanism by which 'shrinking' body sizes might be brought about, the way this is written it appears to be the sole reason, which is not the case. Moreover the evidence for the TSR is currently contradictory due to discrepancies in study methodology:

<https://www.authorea.com/doi/full/10.22541/au.171813253.384000>

3) Mortality could also be a driver, not just changes in growth

A solution could be to mention these other potential drivers: e.g. "due to ecological drivers such as changes in species or trophic guild composition, or physiological drivers such as changes in asymptotic body sizes"

Line 62

I do not think that higher temperatures "favour" smaller species, it is a consequence of temperature. Perhaps: "similar to the observed impacts of increasing temperatures, fishing also results in an increase in the relative abundance of smaller-bodied fish, as commercial fishing....etc". But perhaps in a more concise way.

Line 77 - 79

This sentence is quite convoluted, I wonder if it could be made more clear and concise.

Line 98

This sentence is unclear: you need to state "increasing" or "decreasing" before the thing changing, perhaps reword the sentence to describe the relationship as either positive or negative, whichever suits.

Line 108 - 109

I think this is unlikely to be used in short forecasts - I think the term projection is more appropriate here, e.g. referring to modelling work that supports EBFM and uses PPMR in parameterisation.

Line 113 - 120

This section was very clear and nicely explained, good job.

Line 133 - 134

Before or after accounting for fishing effects? Please clarify here

Line 134

"Predator body mass": Is this the mean or maximum? I am assuming mean, but please restate that. I assume this is without accounting for differences in fishing pressure along the temperature gradient, even if it is equal along the gradient just say "after accounting for fishing pressure"

Line 139

It is unclear to me how predator and prey species were defined and whether there was overlap, so this is difficult to interpret.

Line 143

I would not put "physiological responses" here, there could be other mechanisms at play. No need to draw conclusions, just say "intraspecific responses" then if you want to discuss physiological mechanisms bring that explanation into the discussion.

Line 146

Before or after or without accounting for fishing pressure?

Line 177-178

This result is why it would be helpful to clarify 'before accounting for fishing pressure' in the above. Is there a relationship between fishing pressure and temperature across your sites? If not this would be helpful to know.

Line 196

Figure 3: If you made the geom_points slightly more transparent, or the lines layered on top of them a slightly darker shade of their current colours, it may be easier to see the lines. As it is the 'large prey size' at 'low fishing' in 'panel C' is very hard

to see.

Moreover, it is confusing that you use a solid line to represent the small prey size class in panel C, when you use a solid line elsewhere to represent other things. Perhaps use a dotted line instead for panel C. Keep using the solid lines as you do in the other plots (where they do not specifically relate to small prey).

Panel d: was commercial fishing only analysed as a continuous variable for this one test and elsewhere it was categorical? You also only indicate low and high in the figure, but in the caption describe a 'medium' level?

I find the part where you describe the trendlines confusing, especially as it comes before describing panel e, perhaps describe this where you describe each panel.

Panel e: you have "prey weight class" and "prey body mass" both on this figure. How are they different? Is one species-based? Is one the binned version of the other?

Line 221

Instead of just "climate change" can you be more specific and say "increased water temperature"?

Line 226

You don't present time series data though, so rather than saying your study provides "information on the historical variation" you might better say your study provides "potential insights into past variation"

Line 228-229

“, given the long timescales over which ecosystems have been exposed to these conditions .” I am not sure what this means and you do not need it.

Line 231

Instead of "wholesale" perhaps use "overall"?

Line 236

Possibly "increase" instead of "increased"

Line 239 -242

This is the Temperature Size Rule (TSR) for which there is contradictory evidence.

Line 242-244

I would not say "more easily" carte blanche. Arguably, larger individuals thermoregulate more easily in cooler conditions, and less easily in warmer conditions – so how easily thermoregulation occurs for a given body size depends on the conditions and this is key. This is stated throughout the cited paper. You could say, "Furthermore, when close to the upper limits of their thermal ranges, smaller individuals thermoregulate more efficiently than larger individuals as they are better able to lose excess heat due to their larger SA:V (citation). This has led to many authors predicting a general decrease in the body sizes of ectotherms with climate-driven warming (citation)".

Line 249

The relationship between PPMR and size-spectrum slopes also varies with temperature and may be driven by compositional and structural shifts within ecosystems with temperature. No need to cite, but this may be of interest: Community size structure varies with predator–prey size relationships and temperature across Australian reefs. Coghlan et al. (2022) *Ecology and Evolution*, Volume 12, Issue 4 e8789.

Line 253

"the distribution of small and large taxa was still similar" in trying to interpret this sentence I am wondering if the mean size of prey items remained the same as a result. If this is so, it would be good to add here, if not, this is also valuable information.

Line 251

“. In other words, while the taxonomic composition of prey may have changed from colder to warmer waters, the distribution of small and large taxa was still similar. The sampled predators did respond to the community-wide intraspecific decrease in prey body mass at higher temperatures and fishing effort by selecting the largest individuals available to them”

These sentences seem contradictory. What specifically do you mean by 'distribution,' is this a size distribution? Does the prey minimum and maximum size range change? It may be valuable to describe how the mean size changed without the size distribution changing.

My current understanding from this sentence is that the taxonomic composition changed, the size distribution remained the same, yet the mean size decreased.

I think this needs to be laid out more clearly for the reader.

Line 257

What do you mean by "increased importance" specifically? Do you mean "increased proportion"? "Importance" is vague. I feel like this paper would benefit from a conceptual diagram showing what is happening in terms of prey availability in the environment, versus what is happening in the diet. There is a lot going on with changing prey size, taxonomic composition, size distributions, and what is being observed in a generalised predator's diet. It is a really challenging result to summarise, I do not envy the task! But it is very valuable work and when presented more clearly will be a very valuable paper in the field. Being very clear and precise in the language used and the results described, even to the point of repetition, will really help make these results accessible to readers. Referring to figures when describing the results will also be very useful and I am surprised this was not done in this section. Even if some figures are supplementary, they will no doubt help with clarity and should be referred to.

Line 266

“Driver” instead of “force”? Optional

Line 269 - 271

“which underpinned the increase in PPMR in areas of high commercial fishing” before or after accounting for changes in temperature?

Line 289

Prey species or prey individuals? This needs to be clarified in every sentence involving the word prey as you could be referring to one or the other, you discuss both in the paper. Here it isn't clear which

Line 289

“regardless of temperature...” this is a confusing sentence, so helps to reiterate, if this is indeed what you mean

Line 290-291

I am not sure what this sentence means, or how it would cause larger prey body sizes. Please elaborate or remove. Perhaps lead with the following hypothesis as that seems to have a bit more solidity (you cite and explain it better). What do you mean by “catch well”?

Line 291 - 295

I think what this (“top down”) means is “the selective removal of the largest individuals in the community may release large prey individuals from the risk of predation, should they outgrow the gape-limitations of their predators [Mixhitalis or someone], or become more costly to catch (larger fish have faster swim speeds [cite] and are more likely to escape predators [cite]).” I think you could afford to elaborate a little more here.

If you mean some other form of top-down control, I think that could also be explained. This is if you are talking about individuals not species, which has not been clarified. If species, you need to elaborate why/how you think top-down control could be manifesting, rather than vaguely refer to it.

Line 293

“The latter” is ambiguous. If you mean “top-down control” please just repeat this, it really helps with clarity.

Line 296

“community-wide reduction in individual prey body mass” Is this the mean or maximum prey body mass? Please be specific.

Line 298

Above you say that there was “reduced predation on larger prey species” here you say there is increased predation on them? I suspect I am missing something, which may sbe due to this paragraph being a little unclear.

Line 302

“with no change in body mass across the temperature gradient” even after accounting for changes in fishing pressure? Or with no change in “temperature or fishing pressure”? It's important to know what the other variable is doing because you mention that there is an interaction.

Lines 302 - 305:

What does this mean? Please explain the sampling bias.

Could it be due to, specifically, the trawls “allowing individuals of the largest body size to escape” or something? What do you mean by “no control on the prey composition included”, perhaps, “some prey are more likely to escape the trawl, such as [some fish] which burrow into the sand at the trawls approach. Should [these species] comprise more of the prey in certain areas, we would miss [this component] in our sampling. However, previous [BRUV or something] surveys have shown the abundance of [these prey] is the same throughout our sampling region, so we don't think this potential bias is likely to impact our findings...” or something. I do not imagine this is actually a bias you faced as I am unfamiliar with your ecosystem, but hopefully you can see how the potential bias is described and then combated. This can be brief but you do need to elaborate on it.

Alternatively, perhaps you mean: “this could be a consequence of the selective removal of [species] from this part of the region where they are commercially valuable (being not targeted elsewhere). Should these species be larger in this area, [the trend we observed] may be masked by the impacts of fishing.”

Or “we expect [the species] targeted by fishing to be more abundant/larger in the heavily fished areas, so it is possible that a compensatory effect is occurring, masking the impact of temperature”. Or something along those lines.

Line 305 -

Above you say there was no change, I believe you mean, “regardless of fishing pressure there was no change with temperature” up there, I'll make a note, but this needs to be clarified each time. You know this data back-to-front and it is a great analysis on an amazing dataset, but it has the potential to be very confusing for the reader if things are not reiterated and clarified. Some of these things are: whether you controlled for (or ignoring) one factor or another (e.g. temperature or fisheries”, whether you are talking about prey species or individuals, and reiterating you are talking about the community [mean] body size or the species [max] body size, etc. where appropriate.

Line 307-309 -

Here you clarify that there is an interaction effect. Perhaps state this as the first sentence above. “Due to the interaction effect between fishing and temperature, we cannot interpret the effects of temperature on the body size etc without accounting for

shifts in fishing pressure.” Be super clear from the go.

Line 309-312 -

“The original hypothesis whereby PPMR would increase in the Northeast Atlantic due to increased temperature was only substantiated by a decrease in prey body mass, not predator body mass as anticipated”.
This is a key result. I feel like this could be stated earlier and have its own paragraph.

Line 312 - An alternative, perhaps more clear way to say this:

However, there was an interaction effect between pred body size and fisheries impact, such that when fisheries was included we did see the anticipated reduction in pred body mass with temperature”

Line 314-316 - This sentence sort of comes out of no-where and could be better off in the conclusion.

Line 317 - 319 - Change “An increase” to “Increasing” and “can be associated” to “is” and the language becomes more active, plus you save words

Line 319-320 - I found this sentence very confusing, what is “absolute trophic energy”? Do you mean fewer steps in the food chain, thus less energy lost through “trophic transfer inefficiencies”?

Line 320-322 - Elaborate, i.e., flexible foraging or generalised predation can prevent the over-dominance or over-predation of particular prey species under different environmental conditions, leading to imbalances in ecosystem structuring.

Line 322 - As you are expanding on the above sentence, do not use “furthermore” as it suggests to the reader that you are introducing a new, additional point.

Line 324 - This is essentially what I suggested you write two comments above, but with more jargon. It is along the lines of what I suggested, but you introduce “secondary extinctions” and then do not going into it, so perhaps remove that and keep it simple. Also try to keep relevant sentences together in the paragraph to avoid jumping around.

Line 331 - What does this mean? Lower energy efficiency causes weak interactions? Are you referring to trophic transfer efficiency? Because that is an artifact of weak interactions and not a cause. Perhaps you mean “Given energetic variation in prey, generalised predation may lead to predators selectively targeting larger individuals (of any prey species) in order to ensure energetic requirements are met per feeding attempt”

Line 335 - 337 - This sentence is unexpected, I am not sure whether this paragraph is discussing trophic transfer efficiency, or optimal foraging, because it seems to do a bit of both. Perhaps separate paragraphs for those two topics?

Line 339 -I would not say that it constrains, rather predicts

Line 339 It can also be used to estimate the community size spectra
<https://onlinelibrary.wiley.com/doi/full/10.1002/ece3.8789>

Reviewer #3

(Remarks to the Author)

Version 1:

Reviewer comments:

Reviewer #1

(Remarks to the Author)

I have been reading the revision of this manuscript with great pleasure. I have seen that all my points have been addressed in a very thorough manner and I highly appreciate the high scientific standard of the response and the revision. In my view, the comments by all reviewers have helped to strengthen the conclusion of the manuscript. I have thus no further remarks and I am sure that this will be a very impactful publication.

*** Note that original reviewers' comments are in bold. Our responses are numbered sequentially, written in plain font, and contain quotations to the revised text in italics with line numbers corresponding to the revised version of the manuscript ***

Reviewer #1 (Remarks to the Author):

I have very much enjoyed reading the manuscript addressing how the interaction between gradients in temperature and intensive fishery affect the predator-prey body-mass ratios of fish predators. The question targeted is highly relevant and considers the interplay between various anthropogenic stressors. The manuscript is well written and the methodological and statistical procedures are presented with all detail necessary to understand or replicate the study. Most of my initial points, after giving the manuscript a quick read, such as effects of spatial autocorrelation and non-linearity in the relationships have been covered in the method section. I was highly impressed with the rigor how this analysis has been carried out. Therefore, I have only very few points, and I hope that they are helpful.

Response #1: Thank you for the kind words and positive overall impression of our manuscript. We appreciate the constructive feedback you have given us below and hope we have satisfactorily addressed all of your points.

P1: The study is based on a space for time substitution. The temperature gradient arises from large natural variability in temperature due to latitude, depth, and seasonality (line 49). I would encourage the authors to also test for the effects of the underlying variables (latitude, depth, seasonality). This not meant to replace the statistical tests in the main manuscript but it would be great to see the results as a supplement to understand which of these factors is the main underlying driver.

Response #2: Thank you for this excellent suggestion. We now include plots of PPMR as a function of latitude, season, and depth in our new Figure S1 and we note in the main results that latitude is the major contributor to the temperature effect on PPMR.

Ln109: *“Latitude was a major contributor to the effect of temperature on PPMR, with less contribution of season and water column depth (Fig. S1).”*

Figure S1. Effects of latitude, season, and mean depth on the predator-prey body mass ratio (PPMR). (a) There was a significant increase in \log_{10} PPMR with increasing latitude ($t_{260508} = 27.83$, $p < 0.001$, $R^2 = 0.73$). (b) There was no significant effect of season on \log_{10} PPMR ($F_{3,50433} = 1.88$, $p = 0.171$). (c) There was no significant effect of depth on \log_{10} PPMR ($t_{64886} = 0.92$, $p = 0.361$).

P2: The authors have used mixed models with a variety of random effects. I was wondering why predator identity has not been used as a random effect. Of course, this can only be done for predators with sufficient replication across the temperature gradient but it would be important to show that these are effects of temperature not affected by any turnover in predator identity.

Response #3: There is good replication for all the predator species included in the analysis and so we actually did use predator species identity as a random effect in our mixed effects models. We made the oversight of omitting stomach identity (i.e. the identity of individual predators) from the random effects structure though, so have now updated this in the revised manuscript. We think this helps to account for the non-independence of multiple prey sizes in the stomach contents of each individual predator.

Ln435: “The model included the fixed effect of mean temperature ($^{\circ}\text{C}$, continuous variable) and random effects for chlorophyll ($\mu\text{g/l}$, continuous), salinity (‰ , continuous), depth (m, continuous), number of years since the start of the study period (continuous), ICES rectangle (categorical), season (categorical), predator species identity (categorical), and predator stomach identity (categorical).”

P3: I like how the results are presented in the figures. Nevertheless, I would like to get a better feel for the quantitative effects. I would like to see some numbers on what is the predicted average change in PPMR on the temperature gradient between five and twenty degrees Celsius.

Response #4: This is a great suggestion thank you. The predicted mean increase in PPMR across the temperature gradient without accounting for commercial fishing was 30 %, but it was 22 % in areas of low commercial fishing and 82 % in areas of high commercial fishing. We added this information to the revised manuscript.

Ln107: “PPMR was predicted to increase by 30 % across the temperature gradient, or 1.8 % per 1°C .”

Ln115: “In areas of low commercial fishing, PPMR was predicted to increase by 22 % across the temperature gradient, or 1.3 % per 1°C , whereas in areas of high commercial fishing, PPMR was predicted to increase by 82 % across the temperature gradient, or 4.8 % per 1°C .”

P4: In Fig 1d, the interaction effect and the model predict that intensive fishery increases PPMR at high and decreases it at low temperatures. Is this reality or a model artefact? If this is a model artefact, a non-linear model might do a better job despite not being favored by AIC selection.

Response #5: We actually did test for non-linearity in all our models, using AIC to compare between a linear or polynomial fitting. After updating our models to include a random effect for predator stomach identity, we find that the linear model was the better fit in 7 of 10 cases (including the key interactive effect of temperature and commercial fishing on PPMR). We now include a table of AIC comparisons in the supporting information.

Table S3: The AIC values of all linear and polynomial models tested within this study. The optimal model (AIC value > 2 units lower) is highlighted in bold in each case.

Model	Response variable	Linear	Polynomial
PPMR ~ temperature	PPMR	254903.3	255022.9
	Predator mass	5301.242	5308.293
	Prey mass	143835.6	143831.4
	Prey count	114131.7	468990.4
	Prey richness	-85829.94	-86088.3
PPMR ~ temperature × fishing effort	PPMR	246904.1	246939.1
	Predator mass	10427.64	29361.44
	Prey mass	137925.1	137996.6
	Prey count	113464.7	468703.9
	Prey richness	-80542.45	-80906.26

A minor point: line 103: I would not call a change in body mass a physiological effect.

Response #6: We have updated the phrasing to indicate that the physiological process underpinning a change in body mass might be altered metabolic rate.

Ln95: “...increases in PPMR are related to both physiological processes (e.g. altered metabolic rate, which may underpin changes in body mass) and compositional processes (changes in taxonomic composition).”

I hope that these points are useful. As always, I have signed my review and I am happy to discuss these points if anything should be unclear. Signed Ulrich Brose

Response #7: Again, many thanks for your constructive review and taking the time to read over our work.

Reviewer #2 (Remarks to the Author):

The authors leverage an outstanding dataset, probably the most extensive of its kind globally, to investigate a key parameter of size-spectrum models: predator-prey mass ratios. This comprehensive data encompasses both temperature and fishing-pressure gradients, allowing for two potential drivers of fish body size variation in the region to be assessed. Key results include: There is a significant increase in PPMR with increasing temperature (regardless of fishing pressure). The authors next state that there is an interaction effect between temperature and fishing pressure which amplifies the increase in PPMR with temperature. The mean size of predators was not related to temperature (regardless of fishing pressure). Temperature was significantly related to the mean prey size, and interrelated variables: prey count and prey species richness. The above findings are based on mixed effects models of five response variables (i) log₁₀(PPMR), (ii) predator body mass (g), (iii) prey body mass (g), (iv) prey abundance per predator stomach (prey count), and (v) prey species richness per predator stomach. The predictor variables included: (1) fixed effect of mean temperature (°C), random slopes of (2) chlorophyll, (3) salinity, (4) depth, along with random intercepts of (5) stomach contents data source, (6) year since start of study period, (7) ICES rectangle, (8) predator species

identity. These response variables were then re-tested in a model that included the interaction between temperature and commercial fishing effort. The authors state that polynomial and linear models were compared using AIC, and that linear models consistently outperformed polynomial alternatives. While the authors use the well-established approach of linear mixed effects to study the space-for-time patterns of the above variables, there are a few areas that the authors could improve. Mostly these are related to demonstrating the validity of their results and careful interpretation.

Response #8: Thank you for the comprehensive summary of our key findings and the tremendous care and attention with which you have reviewed our paper. We appreciate all the helpful comments and suggestions, which we have taken on board in our revision. Please see our detailed responses to each point below.

The dataset they draw on is publicly available, but the code used for analyses to derive the results was not, this would be beneficial for others and would help with repeatability, transparency, interpretation, and for testing the generality of their results. Also which R package was used for these analyses?

Response #9: We wholeheartedly agree, and it was our attention to upload all R code and associated datasets to the publicly available University of Essex Research Data Repository upon acceptance of the article, as we have done with all our recent publications. We have made them available as supplementary files for review if it is helpful to see them sooner. It should be noted that some models require high computing power due to the size of the dataset.

Greater clarity of the data and statistical model structure - is each data point an individual predator-prey link? It is not always clear if the authors are referring to the mean individual across all species sampled or the species - often the term “individual species” is used, which is confusing and misleading given the random effect of species in the model. The flowchart in the supplement is helpful but would be improved with these additional details. Also, it should be explained in the methods what the later shown statistics are being used to test for. For example, Table S1 is provided to show that predator and prey body sizes do not significantly change with temperature - at the species level, again the nesting in the different tests is not fully clear and should be explicit. Table S1 seems to show predator species not prey as specified in the caption and text. In a size-based view of the world, predators are also prey depending on body size.

Response #10: Every data point is an observation of prey size in the diet of a predator, meaning that there can be multiple PPMR values for each predator if it contains multiple prey individuals in its diet. We now note this in the flowchart (now Figure S2). We have also updated our models to include a random effect for predator stomach identity on the advice of Reviewer #1 (see Response #3), which did not result in any major changes to the patterns described in the original manuscript. Table S1 did actually contain temperature-size responses for both predator and prey, with the heading for prey found further down the table. To avoid confusion, we have now split this into two tables: one for predators and one for prey. We also now more clearly define what we mean by predators and prey at the first relevant point in the results.

Ln140: *“Note that we refer to “prey” here as any organism found in the diet of the fish predators, even if those species could be predators themselves.”*

Predictor variables: Was fullness of stomach considered or level of digestibility? (level of digestibility can make prey sizes inaccurate). Were the prey types considered only fish (as is often the case) or all measured prey (e.g. invertebrates)?

Response #11: All prey types were measured, but fullness of stomach or digestibility were not considered in the database. We have added a comment on these points to the methods.

Ln377: *“All prey species were considered (i.e. both fish and invertebrates). Fullness of stomach or level of digestibility were not considered and so the estimates of body mass may be subject to some associated uncertainty.”*

The spatial pattern of high, medium, and low categories of fishing effort used for visualisation (Figure 1) seems almost random - how were these different levels of effort determined (were they based on quantiles or a cluster analysis or just arbitrarily chosen?).

Response #12: The fishing effort data are not normally distributed and so dividing them into three evenly sized bins would have led to a bias in the number of observations in each category. Accordingly, the fishing data was divided into three categories with an equal number of observations in each bin and then a median value of fishing effort was calculated from each bin. We have added an explanation into the methods.

Ln424: *“For visualisation purposes, the fishing effort data was divided into three bins each containing an equal number of observations, and the median of each bin was calculated for fitting regression lines to plots.”*

It would be helpful to show the residual/diagnostic plots of the model in the supplement. The model predictions shown in Figure 3b at low temperatures and high fishing don't really have any data points, and this is the pattern that drives the interaction term. Appreciate that polynomials were tested, how was model selection carried out? Seeing diagnostics for the mixed effects models would help (e.g. recommendations here: <https://besjournals.onlinelibrary.wiley.com/doi/full/10.1111/2041-210X.12577>).

Response #13: We have added diagnostic plots for the two main models (our new Figures S3 and S4) to show that they conform to the assumptions of normality, homogeneity, and independence of residuals, with no obvious patterns observed against any of the covariates (see example from Figure S3 below). We feel it may be overkill to add diagnostic plots for every model (an additional eight supporting figures with ten panels each, which may overwhelm readers), but we are happy to include these if the editor recommends it.

Figure S3: Diagnostic plots of the model for PPMR vs SST (Figure 1B). (a) Histogram of the residuals, (b) scatterplot of fitted vs residual values, and (c-k) scatterplots of residuals vs each covariate in the model.

Also, please show results in supplement to support this statement: “For all models, linear and polynomial versions were compared using Akaike’s Information Criterion (AIC), but the linear models were consistently found to be the best fit and thus are presented here.”

Response #14: We have included a new Table S3 containing AIC comparisons of all linear and polynomial models (see Response #5). After including predator stomach identity as a random effect, the polynomial model was a better fit in 3 out of 10 cases. We have included this fitting in all relevant figures and updated our results text accordingly. The overall story remains virtually unchanged (though see Response #58 for one minor update).

Overall, while the paper is well-presented, with potentially wide-reaching findings, I find it difficult to assess the validity of these findings in the present version, and therefore whether commercial fishing is really amplifying the PPMR - temperature relationship, without these further details.

Response #15: We hope our changes in the revised manuscript have increased confidence in the results that we present. Thank you for helping to increase the robustness of our study.

More specific comments

Line 15 – 17. The abstract is very well written, excepting this one line where it is unclear whether the same species are found at warm and colder waters, or whether the sentence is referring to different species. The last part of the sentence would be fine if the first bit of the sentence was clear.

Response #16: We have clarified that smaller prey size occurs both within and across species.

Ln15: “*To compensate for smaller prey (both within and across species) in warmer waters and areas of high fishing...*”

Line 33. I would avoid starting sentences with ‘their’ as it is an ambiguous word, ‘trophic’ could be a better word in this case.

Response #17: Changed as suggested.

Line 34. The use of “lower risk”. I think know what you mean, but this is very vague. I am not sure it would be clear to others, perhaps ‘lower risk of injury or wasted energy’.

Response #18: Changed as suggested.

Line 40. “Super trait”. Master is often used, but I wonder if you are choosing ‘super’ to avoid the use of ‘master’, which is up to you.

Response #19: Changed to “*master*”.

Line 42. Not sure ‘moreover’ is appropriate here (given the sentences leading up to it).

Response #20: Changed to “*for instance*”.

Line 50. This is where ‘moreover’ would be appropriate as it is sort of piling more onto the previous sentences. “However” would be appropriate if the previous sentences were contradictory, which they are not.

Response #21: Our logic was that the preceding sentence noted how marine ecosystems exhibit large natural variation in temperature regimes, but the rapid rate of climate change still pushes them beyond this natural variability. We have joined the two sentences to make the link clearer and replaced “*however*” with “*but*”.

Ln41: “*Marine ecosystems exhibit large natural variability in temperature due to latitude, seasonality, and depth, but climate change is causing relatively rapid increases in sea surface temperatures (SST) beyond this natural variability^{21,22}.*”

Line 51. “natural thermal ranges” of what? Please specify what the natural thermal ranges are of, otherwise it is vague.

Response #22: Changed to “beyond this natural variability” and joined up with the previous sentence to make the connection to what we were talking about clearer (see Response #21).

Line 56 – 57. 1) Not solely, it can also be brought about by species redistributions, see Cheung et al., 2013 and others, and possibly through trophic guild-temperature trends, see Coghlan AR, Blanchard JL, Wotherspoon S, Stuart-Smith RD, Edgar GJ, Barrett N, Audzijonyte A. Mean reef fish body size decreases towards warmer waters. Ecology Letters. 2024 Feb;27(2):e14375.

Response #23: This is an excellent point and actually feeds into our later arguments that changes in body size could be driven by either physiological or compositional mechanisms. We now acknowledge that changes in community composition can also lead to a reduction in body mass.

Ln48: “*Decreasing body mass is recognised as a ubiquitous response to increasing temperature, which can be due to physiological drivers such as increased metabolic demands*

keeping species smaller and the age at maturity lower^{7,29}, or through changes in community composition that favour smaller species^{28,30}.”

2) The TSR is only one mechanism by which ‘shrinking’ body sizes might be brought about, the way this is written it appears to be the sole reason, which is not the case. Moreover the evidence for the TSR is currently contradictory due to discrepancies in study methodology: <https://www.authorea.com/doi/full/10.22541/au.171813253.384000>

Response #24: We cannot access the link you recommended because the page cannot be found. We don’t explicitly mention the TSR here or at any point in the manuscript though. Your suggested change in the previous point (see Response #23) means that we now acknowledge multiple mechanisms for shrinking body mass with increasing temperature, so we think there should be no issue here.

3) Mortality could also be a driver, not just changes in growth. A solution could be to mention these other potential drivers: e.g. “due to ecological drivers such as changes in species or trophic guild composition, or physiological drivers such as changes in asymptotic body sizes”

Response #25: We have integrated a version of this suggestion into the text (see Response #23).

Line 62. I do not think that higher temperatures “favour” smaller species, it is a consequence of temperature. Perhaps: “similar to the observed impacts of increasing temperatures, fishing also results in an increase in the relative abundance of smaller-bodied fish, as commercial fishing....etc”. But perhaps in a more concise way.

Response #26: We have made the recommended change.

Line 77 – 79. This sentence is quite convoluted, I wonder if it could be made more clear and concise.

Response #27: We have streamlined the first two sentences of this paragraph to improve clarity.

Ln71: “To improve our predictive capacity, it is important to first understand the historical variation in PPMR across largescale gradients of temperature and commercial fishing pressure.”

Line 98. This sentence is unclear: you need to state “increasing” or “decreasing” before the thing changing, perhaps reword the sentence to describe the relationship as either positive or negative, whichever suits.

Response #28: We don’t quite understand the confusion because the sentence already included directionality throughout, rather than referring to changes. Nevertheless, we have rephrased the beginning as a negative relationship to hopefully make it clearer.

Ln89: “Given that PPMR has a negative relationship with body mass^{1,42,43}, a reduction in the mean body size of the community at higher temperatures and/or commercial fishing pressure should lead to an increase in PPMR since smaller predators consume relatively smaller prey.”

Line 108 – 109. I think this is unlikely to be used in short forecasts - I think the term projection is more appropriate here, e.g. referring to modelling work that supports EBFM and uses PPMR in parameterisation.

Response #29: We changed “forecasts” to “projections” as recommended.

Line 113 – 120. This section was very clear and nicely explained, good job.

Response #30: Thank you!

Line 133 – 134. Before or after accounting for fishing effects? Please clarify here

Response #31: We have relabelled the section as “*Effects of only temperature on body mass, prey count, and prey richness*”. We have also clarified at the end of the first sentence that these analyses were performed “*using the larger dataset without fishing effort data included*”.

Line 134. “Predator body mass”: Is this the mean or maximum? I am assuming mean, but please restate that. I assume this is without accounting for differences in fishing pressure along the temperature gradient, even if it is equal along the gradient just say “after accounting for fishing pressure”

Response #32: Done.

Line 139. It is unclear to me how predator and prey species were defined and whether there was overlap, so this is difficult to interpret.

Response #33: There was indeed overlap, so we have added a sentence to clarify (Ln140; see Response #10).

Line 143. I would not put “physiological responses” here, there could be other mechanisms at play. No need to draw conclusions, just say “intraspecific responses” then if you want to discuss physiological mechanisms bring that explanation into the discussion.

Response #34: Done.

Line 146. Before or after or without accounting for fishing pressure?

Response #35: We have clarified that this analysis was done “*using the larger dataset without fishing effort data included*”.

Line 177-178. This result is why it would be helpful to clarify ‘before accounting for fishing pressure’ in the above. Is there a relationship between fishing pressure and temperature across your sites? If not this would be helpful to know.

Response #36: And we now do that, as recommended (see Responses #31 and #33).

Line 196. Figure 3: If you made the geom_points slightly more transparent, or the lines layered on top of them a slightly darker shade of their current colours, it may be easier to see the lines. As it is the ‘large prey size’ at ‘low fishing’ in ‘panel C’ is very hard to see.

Response #37: We have made the suggested changes, please see the new Figure 3 below.

Figure 3: The interactive effect of temperature (°C) and commercial fishing on **a**, log₁₀ predator body mass (g), **b**, log₁₀ prey body mass (g), **c**, log₁₀ prey count (number of individuals per stomach) and **d**, log₁₀ species richness (number of species per stomach) in the Northeast Atlantic between 1981 and 2016. Commercial fishing effort was analysed as a continuous variable, but for the purposes of visualisation, low (45 – 566 hours per year), medium (576 – 878 hours per year), and high (903 – 3,479 hours per year) levels of fishing effort are indicated in the figure. Trend lines based on the predicted values of the mixed effects model were plotted for low (**a**: $y = 1.6 + 0.026x$; **b**: $y = -1.1 + 0.0039x$) and high commercial fishing (**a**: $y = 1.8 + 0.0081x$; **b**: $y = 1.9 + 0.003x$). The results of **c**, log₁₀ prey count and **d**, log₁₀ species richness were further categorised into prey weight classes by small (0-0.072 g); medium (0.072 – 1.25 g) and large (1.25 – 2,137 g) prey body mass. (**c**: low fishing effort and small species: $y = 2.5 - 0.14x$; high fishing effort and small species: $y = -1.1 + 0.23x$; low fishing effort and large species: $y = 2.3 - 0.12x$; high fishing effort and large species: $y = -1.1 + 0.25x$; **d**: low fishing effort and small species: $y = 0.67 - 4.3 \times 10^{-17}x - 0.00027x^2$; high fishing effort and small species: $y = 0.59 - 3.5 \times 10^{-17}x + 0.0023x^2$; low fishing effort and large species: $y = 0.69 - 2.3 \times 10^{-17}x + 7 \times 10^{-4}x^2$; high fishing effort and large species: $y = 0.63 - 5.8 \times 10^{-17}x + 0.0033x^2$). **e**, nMDS illustrating the weak effects of temperature and fishing effort on the size of prey taxa, with temperature and body mass vectors orthogonal to one another and a weak correlation between body mass and commercial fishing effort ($R^2 = 0.04$). The colour and size of the points represents the size classes of the prey taxa.

Moreover, it is confusing that you use a solid line to represent the small prey size class in panel C, when you use a solid line elsewhere to represent other things. Perhaps use a dotted line instead for panel C. Keep using the solid lines as you do in the other plots (where they do not specifically relate to small prey).

Response #38: We attempted the dotted line as suggested but found it to be even more confusing due to the number of data points behind the trend line. We also tried different types of dashed lines but found it to be similarly confusing. We believe that leaving a solid line is the best option.

Panel d: was commercial fishing only analysed as a continuous variable for this one test and elsewhere it was categorical? You also only indicate low and high in the figure, but in the caption describe a ‘medium’ level?

Response #39: Commercial fishing effort was always analysed as a continuous variable and we simply use the categories of low, medium, and high effort as a tool to visualise the patterns. We note this in the legend for Figures 1 and 3 (i.e. the same approach was also adopted for the analysis of PPMR). We now also specify in the methods that commercial fishing effort was treated as a continuous variable.

Ln440: “The same five response variables (PPMR, predator body mass, prey body mass, prey count, and prey species richness) were then separately included in models containing main and interactive effects of mean temperature (°C, continuous variable) and mean commercial fishing effort (hours per year, continuous variable) for the subset of data that included information on fishing effort.”

I find the part where you describe the trendlines confusing, especially as it comes before describing panel e, perhaps describe this where you describe each panel.

Response #40: We tried moving the trendline information after each panel letter, but we felt it became even more confusing. We need to first explain that commercial fishing effort was visualised at low, medium, and high levels, and that prey count and richness were further categorised by small, medium, and large prey sizes before presenting the regression lines.

Panel e: you have “prey weight class” and “prey body mass” both on this figure. How are they different? Is one species-based? Is one the binned version of the other?

Response #41: Yes, prey weight class is a binned version of prey body mass, with the specification of bin sizes noted in the figure caption. The prey weight classes were included in the nMDS to visualise any potential groupings by prey weight within the data, but a continuous vector of body masses was fitted to test for the amount of variability in community composition explained by prey body mass (very low in both Figures 2e and 3e).

Line 221. Instead of just “climate change” can you be more specific and say “increased water temperature”?

Response #42: Done.

Line 226. You don’t present time series data though, so rather than saying your study provides “information on the historical variation” you might better say your study provides “potential insights into past variation”

Response #43: We made the suggested change.

Line 228-229. “, given the long timescales over which ecosystems have been exposed to these conditions .” I am not sure what this means and you do not need it.

Response #44: The point is that this is a space-for-time substitution rather than an acute warming experiment. Therefore, ecosystems have been chronically exposed to their given temperature regimes for many decades (and thus generations) of their constituent organisms, which are likely to experience adaptation to those environmental conditions. We have clarified this in the text.

Ln231: “...i.e. organisms may be thermally adapted after many generations at higher temperatures, which is not possible to study in short-term experiments involving acute temperature exposures.”

Line 231. Instead of “wholesale” perhaps use “overall”?

Response #45: Done

Line 236. Possibly “increase” instead of “increased”

Response #46: Changed

Line 239 -242. This is the Temperature Size Rule (TSR) for which there is contradictory evidence.

Response #47: We acknowledge that there can be exceptions, referencing a recent paper by Wootton et al. showing how warming does not have to elevate metabolism if there is sufficient time for acclimation, and that reductions in body size can instead emerge from increased investment into reproduction.

Ln245: “...though note there are exceptions to this general rule⁵².”

Line 242-244. I would not say “more easily” carte blanche. Arguably, larger individuals thermoregulate more easily in cooler conditions, and less easily in warmer conditions – so how easily thermoregulation occurs for a given body size depends on the conditions and this is key. This is stated throughout the cited paper. You could say, “Furthermore, when close to the upper limits of their thermal ranges, smaller individuals thermoregulate more efficiently than larger individuals as they are better able to lose excess heat due to their larger SA:V (citation). This has led to many authors predicting a general decrease in the body sizes of ectotherms with climate-driven warming (citation)”.

Response #48: We agree and really appreciate your description. We have included the first sentence above in the revised manuscript (Ln245).

Line 249. The relationship between PPMR and size-spectrum slopes also varies with temperature and may be driven by compositional and structural shifts within ecosystems with temperature. No need to cite, but this may be of interest: Community size structure varies with predator–prey size relationships and temperature across Australian reefs. Coghlan et al. (2022) Ecology and Evolution, Volume 12, Issue 4 e8789.

Response #49: We are familiar with this paper and now refer to the associated link between PPMR and size spectra at two points in the revised manuscript.

Ln330: “This highlights how temperature can interact with factors like PPMR and the associated indirect changes to trophic interaction strengths to alter entire community size spectra⁷⁷.”

Ln336: “PPMR is a valuable metric to monitor ecosystem changes as it is a good predictor of trophic interactions, governs how energy flows through ecosystems^{1,2,13}, can estimate community size-spectrum⁷⁷ and as a result, is a fundamental input into size-based models of ecosystem dynamics such as the Allometric Diet Breadth Model^{2,3}.”

Line 253. “the distribution of small and large taxa was still similar” in trying to interpret this sentence I am wondering if the mean size of prey items remained the same as a result. If this is so, it would be good to add here, if not, this is also valuable information.

Response #50: We address this comment in more detail in the next response.

Line 251. “. In other words, while the taxonomic composition of prey may have changed from colder to warmer waters, the distribution of small and large taxa was still similar. The sampled predators did respond to the community-wide intraspecific decrease in prey body mass at higher temperatures and fishing effort by selecting the largest individuals available to them”. These sentences seem contradictory. What specifically do you mean by ‘distribution,’ is this a size distribution? Does the prey minimum and maximum size range change? It may be valuable to describe how the mean size changed without the size distribution changing. My current understanding from this sentence is that the taxonomic composition changed, the size distribution remained the same, yet the mean size decreased. I think this needs to be laid out more clearly for the reader.

Response #51: Prey body mass decreased with increasing temperature, but this was driven by systematic within-species reductions in body mass, rather than changes in taxonomic composition that were biased towards smaller taxa (across-species reductions in body mass). We have tried to clarify this by replacing the word “distribution” with “number”, highlighting that the driver of decreasing prey body mass is “within rather than across species” and stressing that there is an “intraspecific decrease in prey body mass”.

Ln256: “In other words, while the taxonomic composition of prey species may have changed from colder to warmer waters, the number of small and large taxa was still similar. Despite the similar size composition of prey species across the temperature gradient, there was a community-wide intraspecific decrease in prey body mass at higher temperatures and fishing effort (Table S2), i.e. driven by reductions in size within rather than across species. The sampled predators responded to the intraspecific decrease in prey body mass by selecting the largest individuals available to them (Fig. 2c) and a greater diversity of large prey (Fig. 2d), which could be a result of optimal foraging in order to maintain energetic requirements⁶⁰.”

Line 257. What do you mean by “increased importance” specifically? Do you mean “increased proportion”? “Importance” is vague. I feel like this paper would benefit from a conceptual diagram showing what is happening in terms of prey availability in the environment, versus what is happening in the diet. There is a lot going on with changing prey size, taxonomic composition, size distributions, and what is being observed in a generalised predator’s diet. It is a really challenging result to summarise, I do not envy the task! But it is very valuable work and when presented more clearly will be a very valuable paper in the field. Being very clear and precise in the language used and the results described, even to the point of repetition, will really help make these results accessible to readers. Referring to figures when describing the results will also be very useful and I am surprised this was not done in this section. Even if some figures are supplementary, they will no doubt help with clarity and should be referred to.

Response #52: We have deleted this sentence in the revised manuscript. Unfortunately, we do not have access to any data on prey availability in the environment, so we cannot produce such a conceptual figure. We do however refer back to key results figures throughout the discussion now, as recommended.

Line 266. “Driver” instead of “force”? Optional

Response #53: Done

Line 269 – 271. “which underpinned the increase in PPMR in areas of high commercial fishing” before or after accounting for changes in temperature?

Response #54: After accounting for temperature. The sentence was changed to reflect that.

Ln275: *“In particular, commercial fishing magnified the reduction in the body mass of prey species with increasing temperature (Fig. 3b), which underpinned the increase in PPMR in warmer waters with more fishing.”*

Line 289. Prey species or prey individuals? This needs to be clarified in every sentence involving the word prey as you could be referring to one or the other, you discuss both in the paper. Here it isn’t clear which

Response #55: We have deleted that sentence in the revised manuscript, but we also reviewed the manuscript with this comment in mind and changed Ln257 as well to refer to “prey species”.

Line 289. “regardless of temperature...” this is a confusing sentence, so helps to reiterate, if this is indeed what you mean

Response #56: This entire paragraph has been deleted in the revised manuscript (see Response #58).

Line 290-291. I am not sure what this sentence means, or how it would cause larger prey body sizes. Please elaborate or remove. Perhaps lead with the following hypothesis as that seems to have a bit more solidity (you cite and explain it better). What do you mean by “catch well”?

Response #57: We acknowledge that this sentence was confusing and so we have now deleted it.

Line 291 – 295. I think what this (“top down”) means is “the selective removal of the largest individuals in the community may release large prey individuals from the risk of predation, should they outgrow the gape-limitations of their predators [Mixhitalis or someone], or become more costly to catch (larger fish have faster swim speeds [cite] and are more likely to escape predators [cite]).” I think you could afford to elaborate a little more here. If you mean some other form of top-down control, I think that could also be explained. This is if you are talking about individuals not species, which has not been clarified. If species, you need to elaborate why/how you think top-down control could be manifesting, rather than vaguely refer to it.

Response #58: In updating our models with predator stomach identity as a random effect, we no longer find that prey size is consistently larger at high levels of fishing effort. Previously,

this result was dominated by a cluster of points around 10 °C, which related to many individual prey measurements in the diet of only a few predator individuals. Our new model structure downweights this bias and provides a result that makes more sense for the greater reduction in PPMR with increasing temperature at high levels of fishing effort. This paragraph on altered top-down control is no longer relevant, so we have deleted it.

Line 293. “The latter” is ambiguous. If you mean “top-down control” please just repeat this, it really helps with clarity.

Response #59: As noted in the previous comment, we have now deleted this paragraph.

Line 296. “community-wide reduction in individual prey body mass” Is this the mean or maximum prey body mass? Please be specific.

Response #60: We have now deleted this sentence.

Line 298. Above you say that there was “reduced predation on larger prey species” here you say there is increased predation on them? I suspect I am missing something, which may sbe due to this paragraph being a little unclear.

Response #61: We agree that the paragraph was a little unclear and we have now deleted it as explained in Response #58.

Line 302. “with no change in body mass across the temperature gradient” even after accounting for changes in fishing pressure? Or with no change in “temperature or fishing pressure”? It’s important to know what the other variable is doing because you mention that there is an interaction.

Response #62: We have updated the text and also refer to appropriate figures to clarify the results we mean.

Ln293: “The sampled Northeast Atlantic predators in this study were seemingly more resilient to increasing temperature, with no change in body mass across the temperature gradient (Fig. 2a) or in response to fishing (Fig. 3a).”

Lines 302 - 305: What does this mean? Please explain the sampling bias. Could it be due to, specifically, the trawls “allowing individuals of the largest body size to escape” or something? What do you mean by “no control on the prey composition included”, perhaps, “some prey are more likely to escape the trawl, such as [some fish] which burrow into the sand at the trawls approach. Should [these species] comprise more of the prey in certain areas, we would miss [this component] in our sampling. However, previous [BRUV or something] surveys have shown the abundance of [these prey] is the same throughout our sampling region, so we don’t think this potential bias is likely to impact our findings...” or something. I do not imagine this is actually a bias you faced as I am unfamiliar with your ecosystem, but hopefully you can see how the potential bias is described and then combated. This can be brief but you do need to elaborate on it. Alternatively, perhaps you mean: “this could be a consequence of the selective removal of [species] from this part of the region where they are commercially valuable (being not targeted elsewhere). Should these species be larger in this area, [the trend we observed] may be masked by the impacts of fishing.” Or “we expect [the species] targeted by fishing to be more abundant/larger in the heavily fished areas, so it is possible that a

compensatory effect is occurring, masking the impact of temperature”. Or something along those lines.

Response #63: We now clarify what we mean by potential biases in the sampling of the predators and the prey in the study.

Ln295: *“The consistency in predator body mass could be due to sampling bias of the survey trawls targeting larger, commercially important fish species, i.e. smaller species not captured by the trawls may also be predators in the wider food web, and so the lack of temperature effect on predator body size is largely constrained to organisms in the 1 to 10,000 g size range. The trawls had no control over the prey composition included, i.e. since the prey in the stomach contents were selectively sampled through feeding by the predators.”*

Line 305 - Above you say there was no change, I believe you mean, “regardless of fishing pressure there was no change with temperature” up there, I’ll make a note, but this needs to be clarified each time. You know this data back-to-front and it is a great analysis on an amazing dataset, but it has the potential to be very confusing for the reader if things are not reiterated and clarified. Some of these things are: whether you controlled for (or ignoring) one factor or another (e.g. temperature or fisheries”, whether you are talking about prey species or individuals, and reiterating you are talking about the community [mean] body size or the species [max] body size, etc. where appropriate.

Response #64: We have done our best to ensure that we clarify throughout whenever we are talking about one or both main explanatory variables, prey species or individuals, and community- or species-level body size. Thank you as well for the suggestion to refer to results figures throughout the discussion, which we think will also help to avoid confusion about our meaning (see Response #52).

Line 307-309 - Here you clarify that there is an interaction effect. Perhaps state this as the first sentence above. “Due to the interaction effect between fishing and temperature, we cannot interpret the effects of temperature on the body size etc without accounting for shifts in fishing pressure.” Be super clear from the go.

Response #65: In addressing Response #62, we now specify at the beginning of the paragraph that there is no effect of temperature or fishing effort, and we refer to the results figures corresponding to our logic at each point. We hope this makes things much clearer now.

Line 309-312 - “The original hypothesis whereby PPMR would increase in the Northeast Atlantic due to increased temperature was only substantiated by a decrease in prey body mass, not predator body mass as anticipated”. This is a key result. I feel like this could be stated earlier and have its own paragraph.

Response #66: We prefer to keep the current order of the discussion, but we have created a new paragraph discussing the importance of this key result and placing it in the context of the wider literature.

Ln302: *“Our original hypothesis that PPMR would increase with temperature in the Northeast Atlantic was underpinned by the expectation of widespread reductions in body mass in warmer waters (i.e. for both predators and prey). Given the observed negative relationship between PPMR and body size^{1,42,43}, a community with a smaller average body mass should thus have a larger PPMR. This hypothesis was only substantiated by a decrease in prey body mass (Fig.*

2b), not also predator body mass as anticipated (Fig. 2a; Fig. 3a). Thus, the overall increase in PPMR with temperature was driven by a different mechanism to our expectation, i.e. contrasting effects of temperature on large and small organisms leading to a divergence in their size ratio. Differential effects of temperature on the body size of trophic groups have recently been described in coastal reef ecosystems, with a similar reduction in the size of smaller fish with increasing temperature and no change in larger piscivores³⁰. This highlights the importance of further research to explore the underlying mechanisms to improve our understanding of how warming may affect different food web compartments and thus overall functioning and stability.”

Line 312 - An alternative, perhaps more clear way to say this: However, there was an interaction effect between pred body size and fisheries impact, such that when fisheries was included we did see the anticipated reduction in pred body mass with temperature”

Response #67: After updating our analyses, there is no longer an interactive effect of temperature and fishing effort on predator body mass, simplifying the conclusions. We now note that predator body mass does not decrease “*across the temperature gradient (Fig. 2a) or in response to fishing (Fig. 3a)*”, citing the corresponding results figure for additional clarity.

Line 314-316 - This sentence sort of comes out of no-where and could be better off in the conclusion.

Response #68: We rewrote this entire section (see Response #66) and so the sentence has now been deleted.

Line 317 - 319 - Change “An increase” to “Increasing” and “can be associated” to “is” and the language becomes more active, plus you save words

Response #69: Thanks, we made these recommended changes.

Line 319-320 - I found this sentence very confusing, what is “absolute trophic energy”? Do you mean fewer steps in the food chain, thus less energy lost through “trophic transfer inefficiencies”?

Response #70: We deleted this fragment and instead explain that weak interactions “*can help to buffer against the destabilising, oscillatory dynamics of strong trophic interactions*⁷³.”

Line 320-322 - Elaborate, i.e., flexible foraging or generalised predation can prevent the over-dominance or over-predation of particular prey species under different environmental conditions, leading to imbalances in ecosystem structuring.

Response #71: We have added the recommended explanation.

Line 322 - As you are expanding on the above sentence, do not use “furthermore” as it suggests to the reader that you are introducing a new, additional point.

Response #72: “*Furthermore*” was deleted from the sentence.

Line 324 - This is essentially what I suggested you write two comments above, but with more jargon. It is along the lines of what I suggested, but you introduce “secondary extinctions” and then do not going into it, so perhaps remove that and keep it simple. Also try to keep relevant sentences together in the paragraph to avoid jumping around.

Response #73: We have replaced this sentence with the text you suggested in Response #71.

Line 331 - What does this mean? Lower energy efficiency causes weak interactions? Are you referring to trophic transfer efficiency? Because that is an artifact of weak interactions and not a cause. Perhaps you mean “Given energetic variation in prey, generalised predation may lead to predators selectively targeting larger individuals (of any prey species) in order to ensure energetic requirements are met per feeding attempt”

Response #74: We realise it was confusing to introduce the concept of energetic efficiency here and so have removed it. The sentence now reads “*On the other hand, weak per capita interactions can also result in species needing to consume more or larger prey to meet their energetic requirements⁶⁰, leading to stronger population-level interaction strengths.*”

Line 335 - 337 - This sentence is unexpected, I am not sure whether this paragraph is discussing trophic transfer efficiency, or optimal foraging, because it seems to do a bit of both. Perhaps separate paragraphs for those two topics?

Response #75: We have removed all reference to energy efficiency in this paragraph to avoid confusion and instead focus our arguments on the implications of altered PPMR for interaction strength and stability. We hope this is clearer.

Line 339 - I would not say that it constrains, rather predicts

Response #76: We would prefer to not use the word “*predict*” as it is already used within the sentence. We instead changed the word “*constrains*” to “*governs*”.

Line 339. It can also be used to estimate the community size spectra <https://onlinelibrary.wiley.com/doi/full/10.1002/ece3.8789>

Response #77: Added, as recommended (Ln330; see Response #49).

Reviewer #3 (Remarks to the Author):

Response #78: Many thanks for supporting this initiative for early career researchers and for your contribution to the very helpful suggestions made as part of this review, which have substantially improved our manuscript.